# An Improved Deep Learning Model for Underwater Species Recognition in Aquaculture

Mahdi Hamzaoui [1,†] , Mohamed Ould-Elhassen Aoueileyine [1,†] , Lamia Romdhani [2,*] and Ridha Bouallegue [1]

1 Innov'COM Laboratory, Higher School of Communication of Tunis (SUPCOM), Technopark Elghazala, Raoued, Ariana 2083, Tunisia; mahdi.hamzaoui@supcom.tn (M.H.); mohamed.ouldelhassen@supcom.tn (M.O.-E.A.); ridha.bouallegue@supcom.tn (R.B.)
2 Core Curriculum Program, Deanship of General Studies, Universityof Qatar, Doha P.O. Box 2713, Qatar
* Correspondence: lamia.romdhani@qu.edu.qa
† These authors contributed equally to this work.

**Abstract:** The ability to differentiate between various fish species plays an essential role in aquaculture. It helps to protect their populations and monitor their health situations and their nutrient systems. However, old machine learning methods are unable to detect objects in images with complex backgrounds and especially in low-light conditions. This paper aims to improve the performance of a YOLO v5 model for fish recognition and classification. In the context of transfer learning, our improved model FishDETECT uses the pre-trained FishMask model. Then it is tested in various complex scenes. The experimental results show that FishDETECT is more effective than a simple YOLO v5 model. Using the evaluation metrics Precision, Recall, and mAP50, our new model achieved accuracy rates of 0.962, 0.978, and 0.995, respectively.

**Keywords:** aquaculture; fish species; computer vision; deep learning; transfer learning

**Key Contribution:** In this study, we present the FishDETECT model, an innovative deep learning architecture specifically designed for underwater species recognition in the context of aquaculture. Our research offers improved accuracy, robustness, efficiency, and generalizability. This innovation has the potential to significantly impact aquaculture practices, research, and conservation efforts, representing a substantial advancement in the field of aquatic species recognition.





## 1. Introduction

Aquaculture has become an elegant research field for its important contribution in global food production and economic growth. Increasing global populations means aquaculture provides a more sustainable and environmentally friendly way of producing animal protein than traditional fishing. In addition, aquaculture offers a viable alternative to wild fish stocks, thereby contributing to food security and sustainable management of aquatic resources [1]. In terms of value, aquaculture production reached a record USD 263.6 billion in 2019, accounting for 48% of the total value of global fish production [2]. In addition, aquaculture can have economic benefits by creating jobs and stimulating local economic development. Approximately 20 million people are employed in the aquaculture sector worldwide [2]. As a result, aquaculture plays a significant role in enhancing food security and alleviating poverty in various regions across the globe.

Conventional aquaculture management typically necessitates direct interaction between the fish farmer and the fish. This stressful method can influence the growth of these species [3,4]. Automated fish species recognition offers numerous advantages. Initially, it enables the more efficient monitoring of fish growth, health, and behavior within the aquatic environment. This proactive approach enables fish farmers to promptly identify and address fish health issues or overpopulation concerns, preventing potential harm to the aquatic ecosystem and fish stocks [5]. Moreover, the automated classification of fish

types facilitates feeding management, allowing for better control over the quantity and timing of food distribution, tailored to the specific fish species and their sizes. This, in turn, enhances feeding efficiency, reduces expenses, and mitigates environmental pollution by averting overfeeding of fish [6,7]. Fish recognition technology holds the potential to significantly contribute to sustainability and environmental conservation by enhancing fish stock management. With the ability to swiftly identify fish species and monitor their development, fish farmers gain deeper insights into the ecological impact of their operations. This knowledge empowers them to make well-informed decisions aimed at preserving the sustainability of their aquaculture production, thereby promoting responsible environmental stewardship [8,9]. Nonetheless, the manual processing of underwater videos and photos proves to be both time-consuming and resource-intensive, incurring substantial expenses. The tracking of fish species is significantly impacted by unrestricted environmental elements such background complexity, brightness, camouflage, and clutter. Its continuous movements and similar shapes make it challenging to distinguish between different types [10]. Therefore, automatic fish detection processing of underwater videos is an excellent alternative [11]. The integration of new technology into the species recognition process has the potential to bring significant enhancements to production [12]. One such promising development is the utilization of computer vision for fish species recognition, a recent technique with the potential to yield several advantages. This includes the rapid detection of fish diseases and various other benefits, which we will discuss further in this study [13,14]. The introduction of deep learning represents a breakthrough in object detection that makes it possible to find objects of various kinds.

This paper presents a study employing advanced deep learning and computer vision techniques to automate the recognition of fish species. A comprehensive overview of previous studies on the same subject, along with their findings, will be discussed in Section 2. Section 4 presents the data collection phase and the data pre-processing methods used in this study. Moreover, we emphasize image segmentation techniques, explain our FishDETECT model's training process, and describe how an embedded device is integrated. Then, in Section 5, we illustrate the study's findings and evaluate the improved model performance over previous approaches. Finally, Section 6 concludes the work with some remarks.

## 2. Related Work

Fish species detection and recognition is an important area of research in aquaculture. It can help the farmer avoid costly and time-consuming human intervention. In this context, many studies have been conducted to recognize fish species using machine learning techniques through several methods. In a recent study, Vikram Deep et al. worked on the fish4knowlge dataset. They applied sharpening operations on the images to improve its qualities. Subsequently, a comparison between several algorithms showed that the hybrid DeepCNN-KNN method is the best, with an accuracy rate that reaches 98.79% [12]. Frank Storbeck et al. acquired length and height data from different locations for each fish type. After training the neural network, the model for recognizing fish types on the basis of their height and length is generated. Specific interventions such as deleting non-useful connections from the nodes were applied to the network architecture. The model achieved an accuracy of 95%. Other fish-specific measurements that are detectable by the camera can be taken in this study. Training the neural networks with the diagonal length, width, and surface area of the fish can improve the fish type recognition task [15]. Many other studies have used the computer vision techniques for fish species classification. Ling Yang et al. developed a computer vision model for fish detection. They have extensively discussed the obstacles encountered, such as similarity of fish colors, high-density scene, darkness, and low resolution. A 3D imaging acquisition system for fish behavior analysis is implemented. The model is trained using deep learning techniques. Other preprocessing techniques can be introduced in this study to remedy the problems associated with the complexity of the scenes. Binarization of the acquired images can give more accurate results than gray image

conversion, for example [10]. For a fish classification based on fish skin, color, and texture features, Jing Hu et al. presented a multi-class support vector machine method (MSVM). Images are captured with a mobile smartphone. Six groups of feature vectors are created. Directed Acyclic Graph Multiple Support Vector Machines (DAGMSVM) is selected in this study as the best classifier with an accuracy ranging from 92.22% to 100% depending on the type of species. Image quality is a key element in this study, as classification is based on fish skin, color, and texture features. A professional camera can improve classification task better than a smartphone camera [16]. Classification using Deep Convolutional Neural Network techniques is very efficient. Praba Hridayami et al. proposed a fish recognition method based on deep convolutional neural networks. The VGG16 model, which has been pre-trained on ImageNet, is used for transfer learning. A dataset of 50 species was used in this study. The CNN model is trained on four different types of image in this dataset: RGB color space image, blending image, canny filter image, and RGB image mixed with blending image. The results of the RGB image mixed with blending image are the best, with an accuracy of 96.4% [17]. Zhiyong Ju et al. built a fish type recognition algorithm based on improved AlexNet model. A new model Fish-AlexNet is developed in this study for use in transfer learning. The improved AlexNet model is trained on a dataset containing 19,717 fish images, which are categorized into 16 classes. This new proposed model has shown a high efficiency [18]. The four deep CNN architectures, VGG16, VGG19, ResNet, and Inception, were used by Anderson Aparecido dos Santos et al. The comparison between VGG16, VGG19, ResNet, and Inception showed that Inception is the best. A new method of fish classification is proposed in this study. It is based on the Inception weights to achieve an accuracy of 87.3%. This method is trained on a dataset of 35 species and 12 fish families [19]. Simegnew Yihunie Alaba et al. proposed a model that starts with object detection using CNN. This is a useful method for the case where more than one fish appears in the image. MobileNetv3-large is used to extract the most important features of the image. These features enable effective fish classification. The results showed that the accuracy of this model reached 80.61% [20]. Image segmentation stands as a crucial technique within the realm of computer vision. This technique plays a pivotal role in pinpointing and recognizing objects present within an image. By breaking down an image into coherent sections, it becomes notably more straightforward to identify particular objects or regions of interest. This method is essential in various tasks, including object identification, object tracking, and understanding the spatial arrangement within a scene. Juan Carlos Ovalle et al. propose a deep learning model for fish recognition and length estimation. The similarity of colors, shapes, and textures of the fish can make identification difficult. To resolve this problem, an image segmentation technique is adapted using the Mask R-CNN algorithm. MobileNet-V is introduced in this work to estimate the fish length. Coco is used as a pre-trained model despite the large volume of dataset used. Fine-tuning techniques are not applied to adapt the pre-trained model to the identification task [21]. YOLO, "You Only Look Once", is a popular method for object detection in computer vision. YOLO is a machine learning algorithm that can detect and classify objects in images or videos in real time. The results of object detection using this method are generally more accurate than the traditional methods. Many studies have used YOLO to solve problems related to object classification in computer vision. Joseph Redmon et al. pre-trained an object detection model on ImageNet-1000 before re-training it with a new model combining YOLO and Fast R-CNN. This study did not focus on detection efficiency with poor-quality scenes [22]. Du Juan compared versions one and two of YOLO with other detection frameworks such as Faster R-CNN, and the results showed that yolo accuracy with YOLO is the highest. This study did not go into the architecture of YOLO to try to improve it [23]. Ref. [24] presents a comprehensive review of single-stage object detectors, specifically YOLOs, regression formulation, their architecture advancements, and performance statistics. Youssef Wageeh et al. combined the two algorithms retinex and YOLO. Retinex is used to improve the quality of images, and YOLO is used to detect fish from images that are already processed, to count their number, and to follow their

paths [25]. A comparison was made between YOLO v2 and Mask R-CNN in the task of detecting the ball in a handball game, showing that Mask R-CNN is more accurate in the detection phase. The YOLO v2 and R-CNN models were pre-trained with the COCO dataset. Despite expanding the dataset with more examples of images from other sports, the accuracy of the models did not exceed 43% using the F1 score metric [26]. The two methods YOLO v3 and Mask R-CNN were combined by Haigang Hu et al. to implement a new method that detects residual feeding in a crab breeding tank, with images acquired in real time. YOLO detects the remaining food and the swimming of the crab. Mask R-CNN performs the segmentation and counts the number of residual foods. Despite the model being improved compared with previous methods, the results show that some residual foods are not detected. Fine-tuning applied to the pre-trained model can improve the detection results [27]. Jin-Hyun Park et al. proposed a method designed with YOLO to detect and count the number of species that are dangerous and can destroy the ecosystem. The proposed method achieves an accuracy of 93.94% and 97.06%, respectively, of two species, Bluegill and Largemouth bass, which are considered to be dangerous fish [28].

## 3. Background

### 3.1. YOLO v5 Algorithm

yolov5 (You Only Look Once version 5) is a deep learning algorithm and architecture used for real-time object detection in images and videos. It is part of the YOLO family of object detection models, which are known for their speed and accuracy in detecting and classifying objects within images and video frames. YOLO models work by dividing an image into a grid and predicting bounding boxes and class probabilities for objects within each grid cell. YOLO v5 is an evolution of the earlier YOLO versions, aiming to improve upon their accuracy and efficiency. The computer vision community has widely adopted YOLO v5 and used it for a variety of purposes, such as surveillance systems, autonomous vehicles, and object detection in images and videos [29].

### 3.2. Image Segmentation with U-Net

In the last 50 years, image segmentation research has yielded ground-breaking discoveries, with classical techniques like as thresholding, pixel clustering, and edge detection [30] serving as the basis of the algorithms. In recent years, new approaches to image processing have emerged. Recently, a wave of artificial neural network algorithms has generated interest. Despite the abundance of literature on segmentation, only a few people are actively involved in data collection. Instead of choosing, the emphasis was on enhancing the accuracy of the segmentation task, as evidenced by research using deep neural networks such as U-net, which is an extension of FCN [31,32] and Mask R-CNN [33]. In this paper, we focus on U-net as a deep learning algorithm in the process of image segmentation. With U-net, we train our CarMask model to generate the mask of the fish found in images.

### 3.3. Transfer Learning

Transfer learning is a machine learning technique where a model developed for a particular task is reused as the starting point for a model on a second task. It leverages the knowledge gained while solving one problem and applies it to a different but related problem. In computer vision, transfer learning is a widely utilized method where a pre-trained model from a huge database is used as a starting point to complete a particular task. The use of a pre-trained model that has already learned general features from a similar task reduces time compared to starting from scratch and training a new model.

### 3.4. Roboflow Platform

Roboflow is a platform designed to manage and preprocess their data for training machine learning models, particularly in the field of computer vision. It focuses on simplifying the process of creating and managing datasets for tasks like image classification and object detection. Roboflow provides a range of tools and features, including annotation and

labeling data, bounding boxes for object detection, segmentation masks, data preprocessing, and data augmentation.

### *3.5. Keras Library*

Keras is an open-source deep learning framework renowned for its user-friendly and intuitive interface. It simplifies the process of building and training neural networks by providing a high-level API that abstracts away much of the complexity associated with deep learning. With Keras, users can easily define and customize neural network architectures, stack different types of layers, specify loss functions and optimization algorithms, and evaluate model performance. Originally designed to work with multiple backend libraries, Keras has become synonymous with TensorFlow, establishing itself as the preferred choice for building and training neural networks within the TensorFlow ecosystem. It boasts an extensive documentation and a wealth of pre-trained models, making it suitable for both beginners and experienced machine learning practitioners seeking an accessible yet powerful tool for deep learning tasks.

## 4. Methodology

### *4.1. Data Collection*

In this work, three datasets were used to improve the performance and accuracy of our FishDETECT fish species detection and classification model:

#### 4.1.1. CarMask Dataset

The CarMask dataset contains more than 4500 images of cars captured from different positions with their appropriate masks. The images in this dataset are of different sizes. It is obtained from the kaggle platform [34].

#### 4.1.2. FishSpecies Dataset

The FishSpecies dataset represents nine different seafood types collected from a supermarket in Izmir, Turkey, for a university–industry collaboration project at Izmir University of Economics, and the work was published in Innovations in Intelligent Systems and Applications Conference, Istanbul, Turkey 2020. As shown in Figure 1, the dataset includes Gilt Head Bream, Red Sea Bream, Sea Bass, Red Mullet, Horse Mackerel, Black Sea Sprat, Striped Red Mullet, Trout, and Shrimp image samples. It contains 9450 images of fish distributed over the different species, as shown in Table 1.

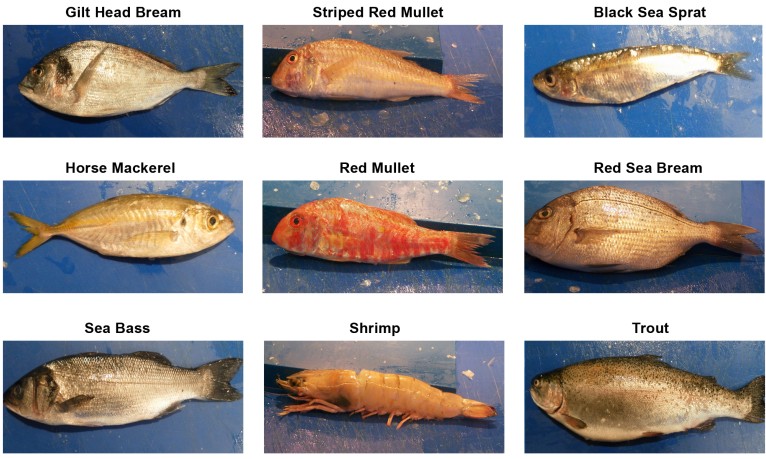

**Figure 1.** Samples of fish images in FishSpecies dataset.

**Table 1.** The label distribution in FishSpecies dataset.

| Fish Species | Count of Samples | Train | Validation | Test |
|---|---|---|---|---|
| Gilt Head Bream | 1200 | 840 | 180 | 180 |
| Red Sea Bream | 900 | 630 | 135 | 135 |
| Sea Bass | 1160 | 812 | 174 | 174 |
| Red Mullet | 900 | 630 | 135 | 135 |
| Horse Mackerel | 1000 | 700 | 150 | 150 |
| Black Sea Sprat | 1150 | 805 | 173 | 172 |
| Striped Red Mullet | 1030 | 721 | 155 | 154 |
| Trout | 1290 | 903 | 194 | 193 |
| Shrimp | 820 | 574 | 123 | 123 |

### 4.1.3. FishMask Dataset

This dataset is built in this work. It contains all the images containing fish masks in the FishSpecies dataset. These masks were predicted using the model for predicting car masks from images of cars after applying fine-tuning.

### 4.2. Data Preprocessing

The Roboflow platform is used for the preprocessing, labelling, and bounding box of both FishSpecies and FishMask datasets. Several preprocessing techniques are exploited, such as Auto-orient, Resize, and Auto-adjust Contrasts. Other data augmentation mechanisms developed by roboflow platform are applied, such as Flip, Rotation, Grayscale, and Brightness. Data augmentation can also reduce overfitting. Image preprocessing procedures are used to improve the model's performances during the training phase. At the end of the task, Roboflow splits the data from each dataset into 3 parts: training, validation, and test. The partitions are adjusted so that 70% of the data are for training, 15% for validation, and 15% for test.

### 4.3. A Deep Learning-Based New Architecture for Underwater Species Recognition

The objective of the architecture proposed in this work is to achieve the most optimal results in the phases of fish object detection and type recognition of these species. As illustrated in Figure 2, the architecture of our system consists mainly of the following elements.

### 4.3.1. CarMask Model

The model is trained on the CarMask dataset using the Keras library. Image segmentation is carried out with the U-net convolutional neural network architecture. It allows us to segment the image for object detection. Subsequently, the model generates the appropriate mask for the found object. After applying fine-tuning to the CarMask model, the latter becomes capable of generating the appropriate fish masks from the FishSpecies dataset. Using the CarMask Model, the FishMask dataset is created. The model takes as input the FishSpecies dataset, which contains fish images, and generates new images containing their predicted masks. The generation of masks helps to recognize the type of fish from its shape.

### 4.3.2. Fine-Tuning Process

Fine-tuning is a technique that involves adjusting a pre-trained model on a specific task using new data while retaining some of the previously learned weights. Firstly, as shown in Figure 3, we began by collecting a few images of fish and creating the appropriate masks manually. The images were annotated so that they could be linked to their masks. Secondly, during the preprocessing phase, all the images were resized to normalize them. Given the small number of images collected, a data augmentation operation was applied to better re-train the model. The previously trained CarMask model is then loaded. The output layer is removed and replaced by another layer suitable for predicting fish masks, whose number of channels is one, since segmentation is binary. The learning rate is lower

than that used in the initial training. After this fine-tuning phase, the model is re-trained and validated. Finally, the CarMask model is generated with its new prediction behavior.

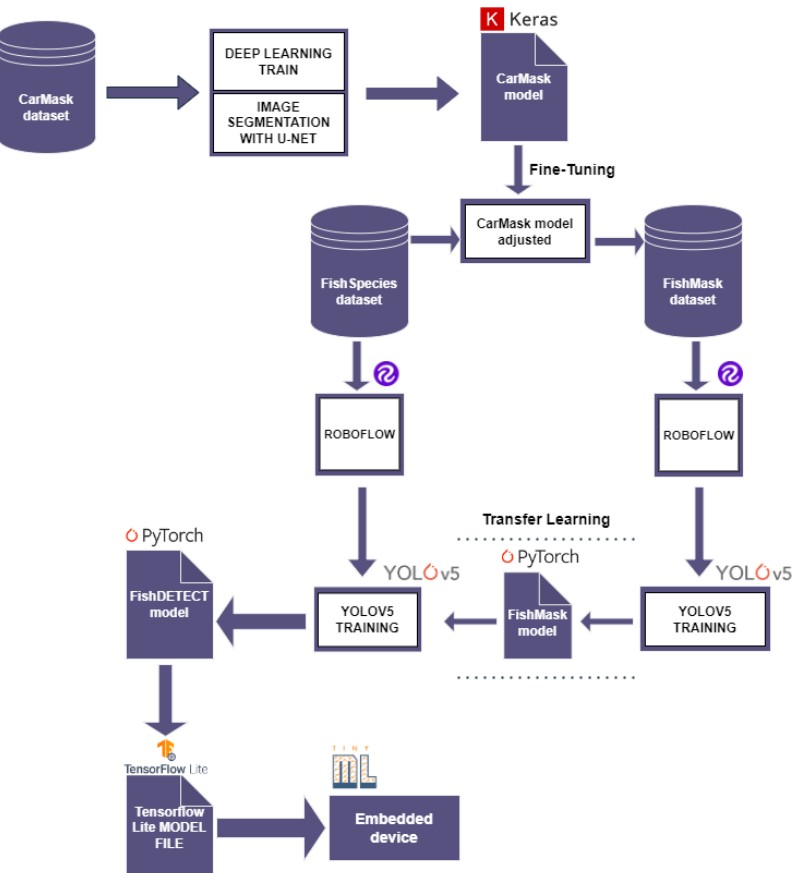

**Figure 2.** The workflow of FishDETECT model.

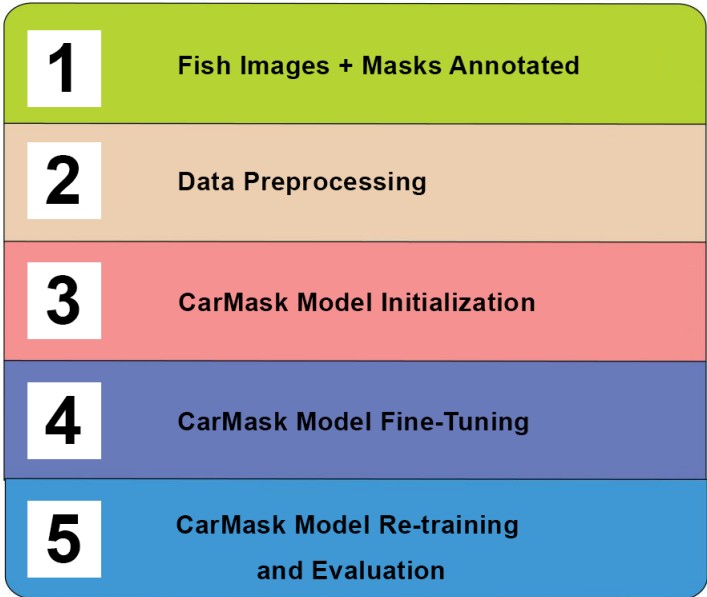

**Figure 3.** The fine-tuning process.

### 4.3.3. FishMask Model

Once the fishMask dataset is created, labeling, preprocessing, and bounding box procedures are applied using the Roboflow platform. The data generated by Roboflow are split into training, test, and validation sets before being used for training with YOLO v5. The model is trained for 30 epochs. Figure 4 shows the training phase of the model. A PyTorch FishMask Model has been created to facilitate the detection of fish objects and automate species recognition from images containing masks. Through the process of transfer learning, the FishMask model enhances the performance of the final FishDETECT model, particularly in environments where distinguishing fish colors is challenging due to darkness.

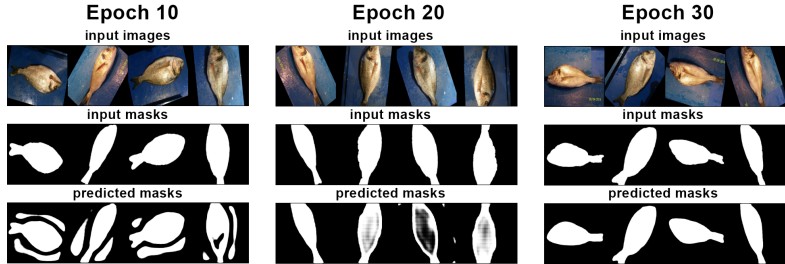

**Figure 4.** Progress of FishMask model training.

### 4.3.4. Our Improved Model FishDETECT

Based on the weights of the previous model, the fish species recognition training is carried out on the FishSpecies dataset using YOLO v5. The labeling, preprocessing, and bounding box annotation are carried out via the Roboflow platform. Subsequently, a PyTorch FishDETECT model is finally ready to be deployed in an embedded environment. To make it easier to use in TinyML, it has been converted into a TensorFlow Lite model in version 2.10.

### 4.3.5. Model Integration into an Embedded Device

In this part, we proceed to the integration of our model in a constrained performance device. We use a Linux-based device and high-resolution camera. We have successfully integrated our FishDETECT model into the device, which utilizes a Raspberry Pi-4 and camera, enabling it to proficiently recognize fish species. This implementation has revolutionized the way we identify and classify various fish species. By harnessing the power of Raspberry Pi-4, a versatile single-board computer, and coupling it with a high-resolution camera, we have created a powerful system capable of capturing and processing images in real time. Through extensive training and fine-tuning of the AI model, the device has acquired the ability to accurately analyze intricate details and distinct features specific to various fish species. Leveraging deep learning algorithms, the embedded AI model efficiently categorizes the captured images, providing valuable insights about the present species. This breakthrough development not only facilitates scientific research and environmental monitoring but also empowers enthusiasts and professionals alike to explore the fascinating world of fish species with enhanced precision and efficiency.

### 4.4. The YOLO v5 FishDETECT Model Architecture

Our FishDETECT model is designed with YOLO v5. As shown in Figure 5, the architecture consists of three important components. The first is the backbone, which is a network that has already been trained to extract a rich representation of image features. This reduces the spatial resolution of the image and increases the resolution of its features. In the current case, the backbone receives fish images in various dimensions before preprocessing. It uses the FishMask Model as a pre-trained model. The second component is the neck. Pyramids of feature extraction are utilized. This makes it easier for the model to generalize the objects of various sizes and scales. The head is the third component that is used to carry

out the last step of the process. It renders the final result, which includes classes, scores, and bounding boxes, by applying anchor boxes to feature maps.

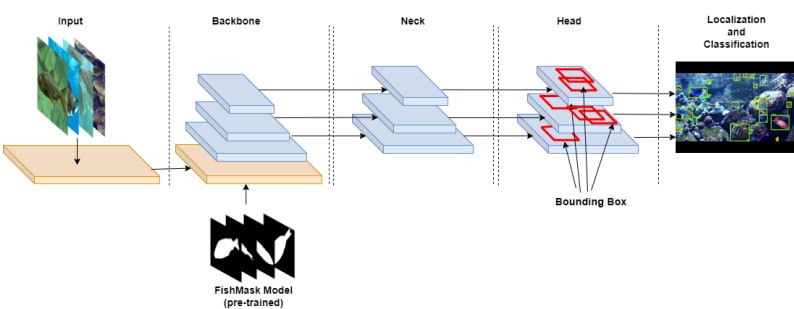

**Figure 5.** The YOLO v5 FishDETECT model architecture.

## 5. Results and Discussion

### 5.1. Pre-Trained Models

In this work, four transfer learning models were passed to our model YOLO v5. The two models, YOLO v5 nano and YOLO v5 large, are two specific variants of YOLO v5 that are represented as a convolutional neural network architecture used for real-time object detection. COCO (Common Objects in Context) is a widely used database for object detection, semantic segmentation, and other computer vision-related tasks. COCO contains over 200,000 images annotated with 80 different object categories, making it a valuable dataset for training and evaluating object detection models. Our pre-trained model is the one we created manually in this study. Problems related to luminosity, scene clutter, and water density influence the quality of the fish images. Training on the fish masks instead of the real images can solve these problems and improve the performance of the model. As shown in Table 2, the performance of the final model is measured individually, with each model run in the context of transfer learning. The model training is carried out over 20 epochs. The performance of the model is measured at the 20th epoch with the Precision, Recall, and mAP50 metrics. The results obtained with YOLO v5 nano are the lowest. The values obtained are 0.832, 0.887, and 0.904, respectively. By applying transfer learning with coco, the model obtains the following values: 0.943, 0.967, and 0.971. YOLO v5 large improves the performance of the model by obtaining the values 0.948, 0.936, and 0.976. Finally, the results obtained with our pre-trained FishMask Model are the best compared to previous pre-trained models. Using the FishMask Model, our final model achieves very high accuracy values. At the 20th epoch, we obtained 0.962, 0.987, and 0.995 for the Precision, Recall, and mAP50 metrics, respectively. The visualization of the result in Figure 6 shows that the training of the FishDETECT model with our own pre-trained model is higher than the training with other existing models. The values of the mAP50 metric are plotted in curves describing the advances obtained in each training cycle.

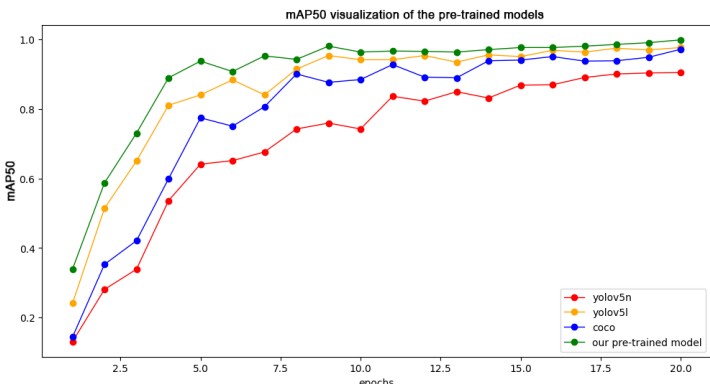

**Figure 6.** The mAP50 visualization of the pre-trained models.

**Table 2.** FishDETECT model evaluation with the pre-trained models.

| Epoch Number | Metrics | YOLO v5 Nano | COCO | YOLO v5 Large | Our Pre-Trained Model |
|---|---|---|---|---|---|
| 1 | Precision | 0.113 | 0.137 | 0.164 | 0.245 |
| | Recall | 0.280 | 0.436 | 0.332 | 0.510 |
| | mAP50 | 0.131 | 0.144 | 0.241 | 0.339 |
| 10 | Precision | 0.742 | 0.832 | 0.895 | 0.912 |
| | Recall | 0.736 | 0.861 | 0.866 | 0.932 |
| | mAP50 | 0.742 | 0.884 | 0.941 | 0.963 |
| 20 | Precision | 0.832 | 0.943 | 0.948 | 0.962 |
| | Recall | 0.887 | 0.967 | 0.936 | 0.978 |
| | mAP50 | 0.904 | 0.971 | 0.976 | 0.995 |

*5.2. FishDETECT Model Performances*

As mentioned in Table 2, the performance of the model is measured with different evaluation metrics. According to the values obtained with the Precision, Recall, and mAP50 metrics, and by transferring the learning from our own pre-trained FishMask model, the model performs well. Figure 7 describes the Precision–Confidence correlation, where Confidence is a numerical value that represents the probability that the detected object is actually present in the image. With a Confidence value of 0.1, the model can achieve an accuracy of 0.9. Gilt Head Bream is the most complicated species in the detection and recognition phase. The detection and classification of Shrimp is the simplest task for the model. The performance of the model is clearly shown in the confusion matrix found in Figure 8. The model achieves very high prediction rates ranging from 0.93 to 1.00. According to the confusion matrix, the model makes some misclassifications between fish species that look similar, such as Gilt Head Bream and Red Sea Bream or Black Sea Sprat and Horse Mackerel. Another slight kind of error is made between the detected fish and the background scene. These small confusions are explained by unclear vision of the scene, low light, poor water quality, and other weather conditions. Data augmentation and the addition of a validation split between the training and test phases solved the problems associated with overfitting. Figure 9 illustrates the training evolution of the model. In each training cycle, our model reduces the loss of object detection in the image.

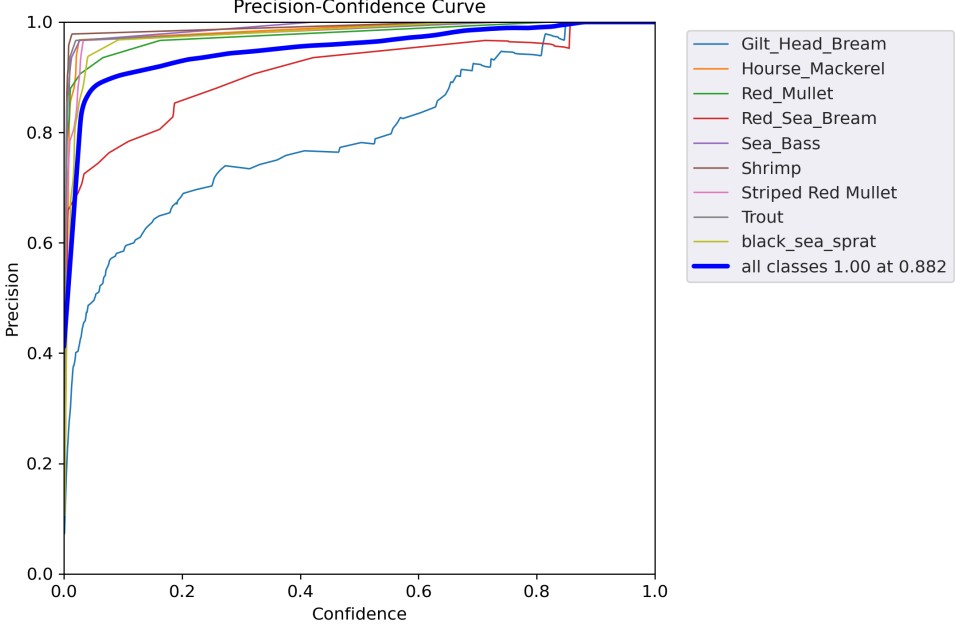

**Figure 7.** The Precision–Confidence curve.

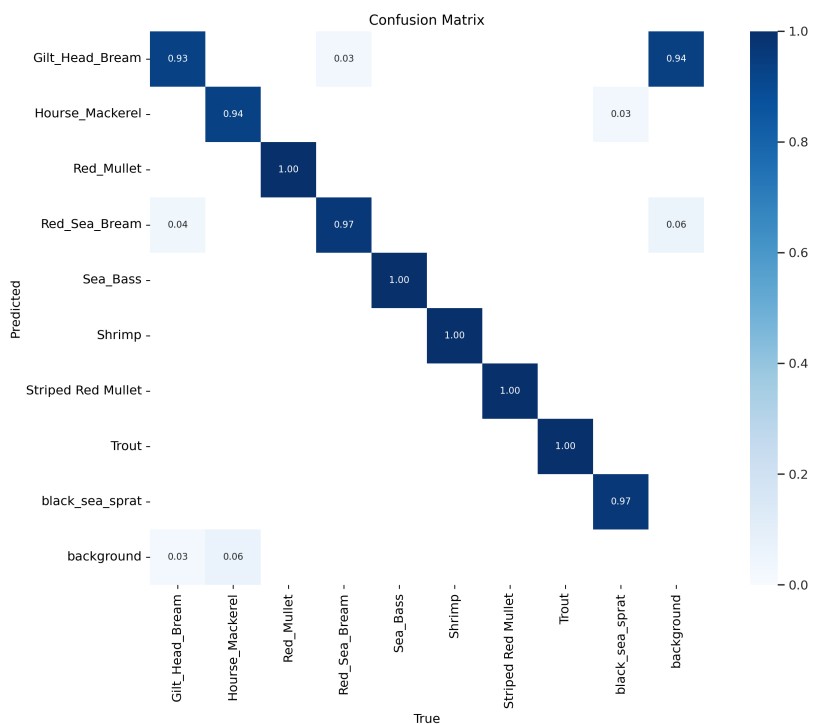

**Figure 8.** The confusion matrix of FishDETECT model.

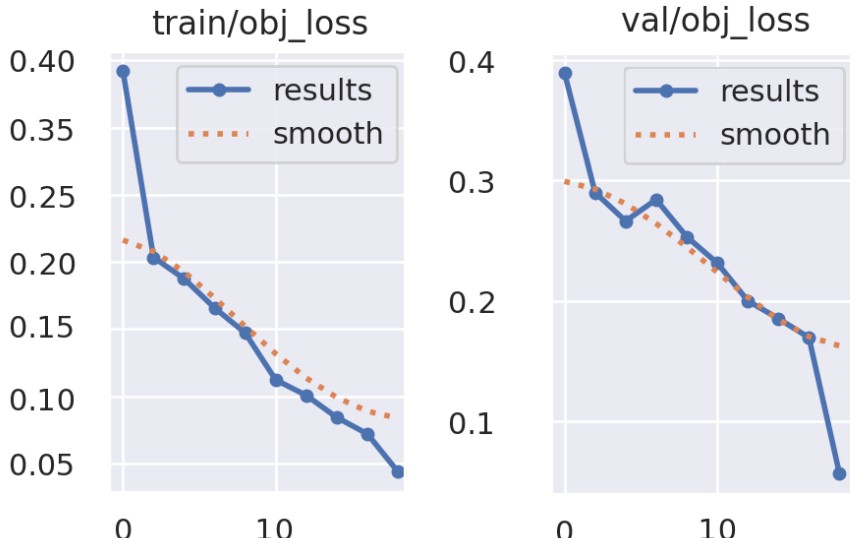

**Figure 9.** The learning evolution of FishDETECT model.

*5.3. Detection and Recognition Results*

In the test phase, underwater videos were captured under different conditions. They were tested with our improved model. Figure 10 shows good-quality videos. The two scenes are captured for the Shrimp and Gilt Head Bream species. As shown in Figure 10, the species detection and classification are of high quality. The errors made by the model are very limited. Figure 11 shows a video taken for a swarm of Trout fish. The conditions are more complex, as the background resembles the color of the fish's skin. As shown in the figure, the luminosity is very low and the water quality seems poor. Despite all these conditions influencing the effectiveness of a computer vision model, our improved model achieved successful detection and classification results.

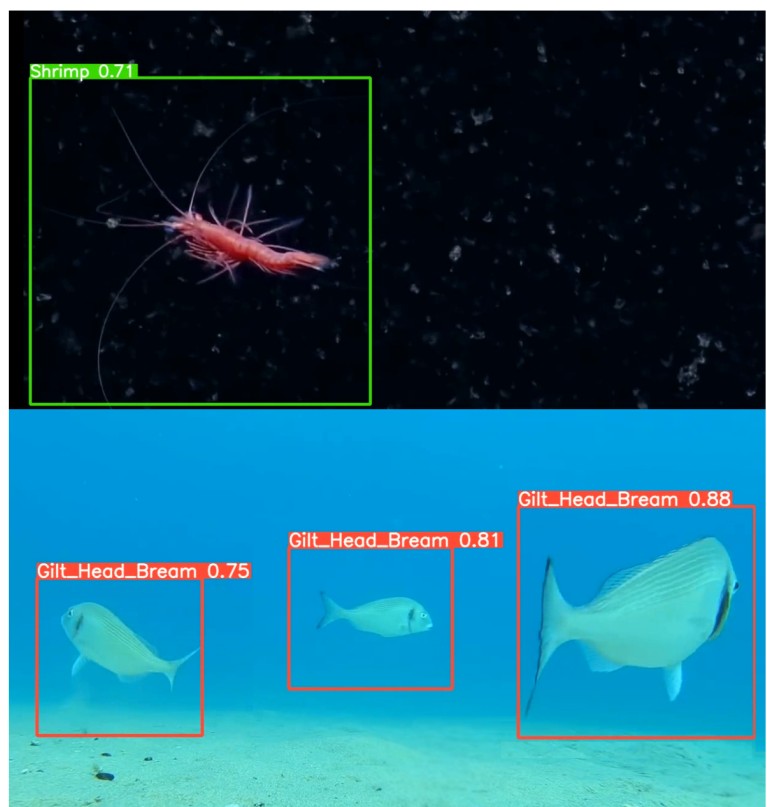

**Figure 10.** Scenes captured in good conditions.

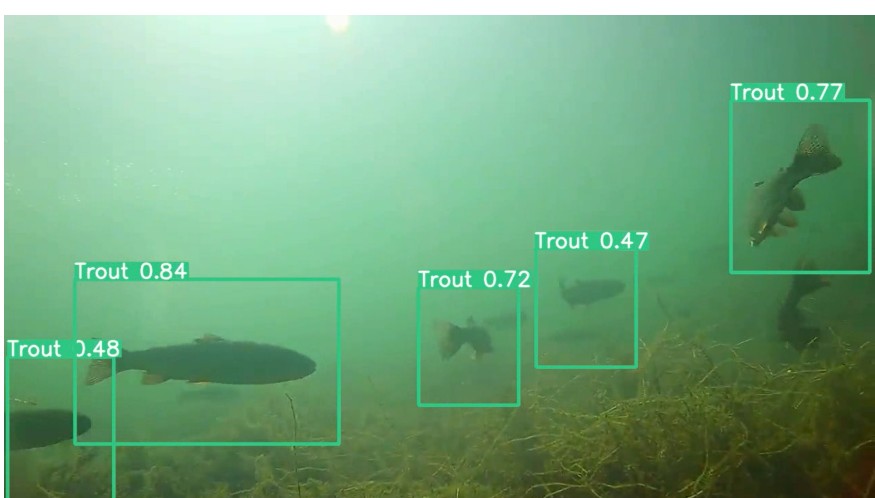

**Figure 11.** Scene captured in poor conditions.

*5.4. FishDETECT Model Integration into the Raspberry Pi-4*

Our model demonstrated its competence by accurately classifying the different types of fish as Trout and Red Mullet with a confidence score of 0.63 and 0.77, respectively. Figure 12 shows the Raspberry Pi-4 board equipped with a camera module. Our FishDETECT model is integrated into this board. Figure 13 illustrates the fish detection phase and automatic recognition of its type in real time. This achievement highlights the potential of our system to make a significant contribution to underwater monitoring and management.

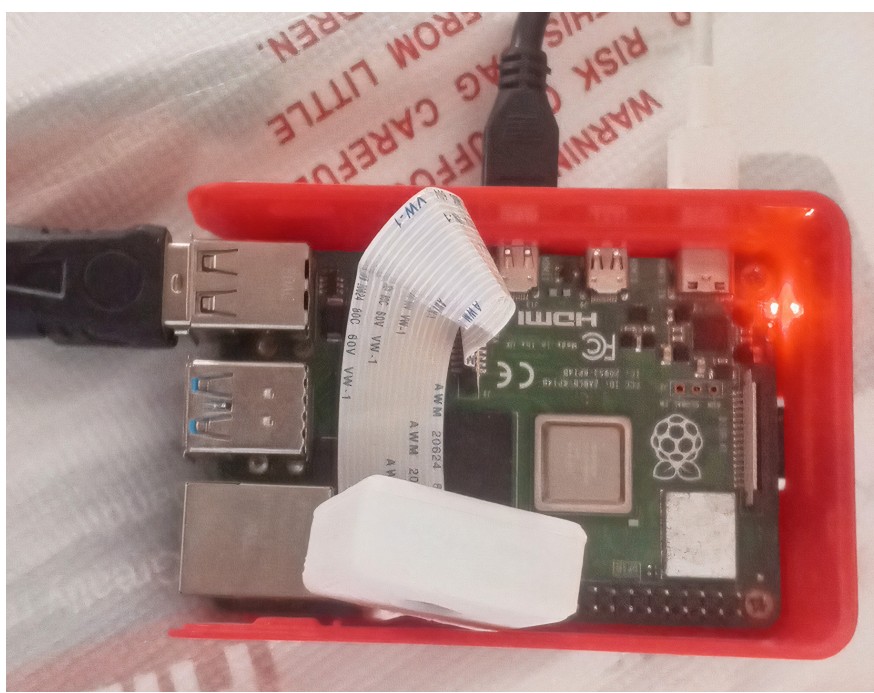

**Figure 12.** Model integration into the Raspberry Pi-4 board.

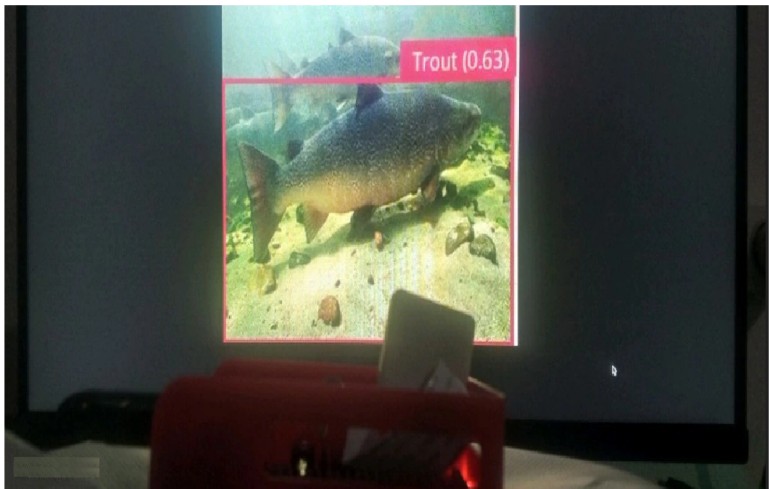

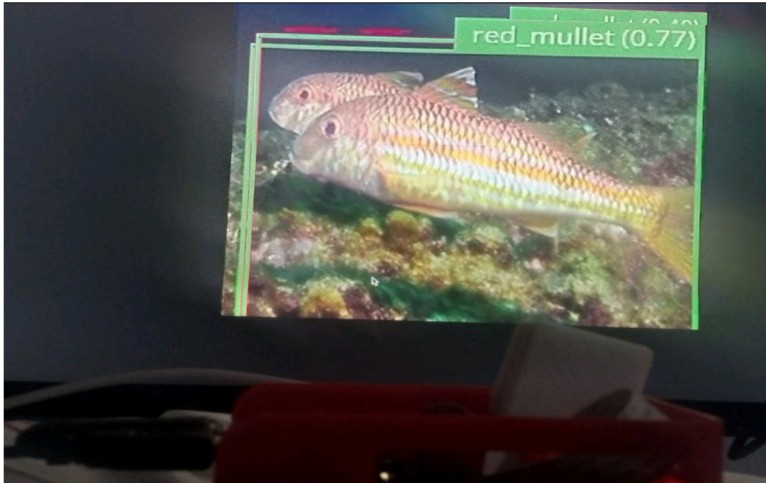

**Figure 13.** Real-time detection visualization.

## 6. Further Discussion

To evaluate the performance of our improved model FishDETECT, the results of this study have been compared with several other similar studies in Table 3. An analysis was made to evaluate the methods used, the data characteristics, the performance indicators, and the results obtained. Based on these comparisons, the advantages and limitations of these fish species recognition methods are elaborated. After comparing our study with other similar work, we can conclude that all these studies detect fish types by applying computer vision techniques. None of the studies used challenges related to environmental conditions such as camouflage, low light, and poor water quality. Our model is broader. It can detect and classify fish species in both good and poor conditions. In addition, the majority of existing studies generally rely on outdated versions of object detection algorithms. However, our work develops a new approach based on YOLO v5 that uses a pre-trained model on fish image masks. This process has enabled us to obtain a preferred model that achieves an accuracy rate of 96.2%. One of the main limitations of our study relates to the quality and quantity of the data used to train the models. The dataset contains biased or incomplete data. To overcome this problem, the use of emerging preprocessing techniques created by the Roboflow platform enabled us to improve the quality and quantity of the data in order to obtain a high-performance model.

**Table 3.** Table of comparisons between this work and related studies.

| Reference | Method | Data Characteristics | Fish Species | Performance Indicators |
|---|---|---|---|---|
| [35] | The method used in this study consists of creating an improved YOLO v3 model that uses the idea of anchor boxes in the prediction phase. The detection scale is raised to 4 instead of 3. To obtain a suitable size of anchor boxes, the K-Means++ algorithm is run with the dataset. Transfer learning took advantage of pre-trained CNN architecture, which had been trained with nearly 1.2 million ImageNet dataset samples and 1000 classes. | The dataset samples were gathered from diverse sources. The dataset's samples are all different sizes, such as $320 \times 320$, $416 \times 416$, and $480 \times 480$. | Anemone-fish. Jelly-fish. Star-fish. Shark. | mAP: 91.30% |
| [36] | This study provides a method based on the YOLO v4 recognition algorithm that has been optimized with a novel labeling technique. | 160 images extracted from videos captured underwater. | Yeesok. Nuanchan. Tapian. Nai. Jeen Ban. Jeen To. Nin. Sawai. | Precision: 98.35% F-Score: 98.96% |
| [37] | An improved real-time detection network was proposed for tuna detection based on the YOLO v3 network, which used lightweight design on the backbone and combined the CBAM attention mechanism module on the basis of the MobileNet v3 network structure to build an efficient tuna detection network, Tuna-YOLO. Following annotation of the dataset, the K-means algorithm was used to obtain nine better anchor boxes based on label information, which was then used to improve detection precision. | All of the image data came from Liancheng Overseas Fishery (Shenzhen) Co., Ltd., and all of the fish were shot on the boat to create catch statistics. | *Xiphias gladius. Thunnus obesus. Thunnus albacares. Makaira mazara.* | Precision: 95.83% mAP50: 85.74% |
| [38] | This study experimented with object detection method based on deep learning, such as Faster R-CNN, which can distinguish the species of fish within an image without additional image preprocessing. | The dataset is obtained from the QUT FISH Dataset. It contains 500 images of 50 classes of fish, with 10 images per class. | *Anyperodon leucogrammicus. Bodianus diana. Cephalopholis sexmaculata. Pseudocheilinus hexataenia.* | Accuracy: 80.4% |

**Table 3.** *Cont.*

| Reference | Method | Data Characteristics | Fish Species | Performance Indicators |
|---|---|---|---|---|
| [39] | This research introduces Composited FishNet, a unique composite fish detection framework based on a composite backbone and an upgraded path aggregation network. A new composite backbone network (CBresnet) is designed to learn scene change information, which is caused by differences in image brightness, fish orientation, seabed structure, aquatic plant movement, fish species shape, and texture differences. | For training, the SeaCLEF 2017 benchmark dataset For training, the SeaCLEF 2017 benchmark dataset is used. This benchmark dataset was created primarily to give resources for evaluating detection algorithms in image and video sequences. The dataset contains 20 low-resolution videos and over 20,000 sample photos of 15 different fish species in their natural coral reef habitat. There are five videos with 640 × 480 pixel resolution and 15 films with 320 × 240 pixel resolution. | 15 different fish species. | Average Precision 0.5:0.95: 75.2% Average Precision 0.5: 92.8% Average Recall: 81.1% |
| This work | The main issue in the field of underwater computer vision is the quality of the image, which is influenced by several factors. Our work consists in developing an improved YOLO v5 model. To detect and classify fish species objects, the model is based on another pre-trained fish masks model instead of using other classical transfer learning sources, such as coco. | The first dataset is called Mask dataset. It contains capture objects of different positions and their appropriate masks. The second is called Fish-Species dataset; it contains images of nine species of fish. The third is FishMask dataset; it consists of the fish masks generated from the second dataset. | Gilt Head Bream. Red Sea Bream. Sea Bass. Red Mullet. Horse Mackerel. Black Sea Sprat. Striped Red Mullet. Trout. Shrimp. | Precision: 0.962 Recall: 0.978 mAP50: 0.995 |

## 7. Conclusions and Future Work

Given the complexity of the underwater environment, the efficiency of the fish detection and classification model is essential. In this paper, we propose a fish recognition solution based on an improved YOLO v5 model called FishDETECT. First, a new transfer learning model called FishMask Model is built. It is trained on a FishMask Dataset containing fish masks and their labels. The dataset is created in this work following the construction of a new model, the CarMask Model. The CarMask Model is trained with the U-Net image segmentation algorithm, which is based on deep learning. This dataset is labeled using the Roboflow platform. FishDETECT model uses the FishMask Model in transfer learning. The results obtained using the Precision, Recall, and mAP50 metrics are more beneficial than those obtained with a simple model based on YOLO v5 nano, YOLO v5 large, or COCO. Several delicate scenes have been tested with our model and we have achieved favorable results. In future work, our aim is to apply computer vision to identify fish diseases in real time. The model will be integrated into an embedded system for immediate interaction between the farmer and his intelligent farm.

**Author Contributions:** Conceptualization, M.H., M.O.-E.A., L.R. and R.B.; methodology, M.H., M.O.-E.A., L.R. and R.B.; software, M.H., M.O.-E.A., L.R. and R.B.; validation, M.H., M.O.-E.A., L.R. and R.B.; formal analysis, M.H., M.O.-E.A., L.R. and R.B.; investigation, M.H., M.O.-E.A., L.R. and R.B.; resources, M.H., M.O.-E.A., L.R. and R.B.; data curation, M.H., M.O.-E.A., L.R. and R.B.; writing—original draft preparation, M.H., M.O.-E.A., L.R. and R.B.; writing—review and editing, M.H., M.O.-E.A., L.R. and R.B.; visualization, M.H., M.O.-E.A., L.R. and R.B.; supervision, M.H., M.O.-E.A., L.R. and R.B.; project administration, L.R. All authors have read and agreed to the published version of the manuscript.

**Funding:** This research received no external funding.

**Institutional Review Board Statement:** Not applicable.

**Informed Consent Statement:** Not applicable.

**Data Availability Statement:** Not applicable.

**Acknowledgments:** Open Access funding provided by the Qatar National Library.

**Conflicts of Interest:** The authors declare no conflict of interest.

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
