# Peer review of "An Improved Deep Learning Model for Underwater Species Recognition in Aquaculture"

_fishes, doi:10.3390/fishes8100514_

Round 1
Reviewer 1 Report
The manuscript titled "An Improved Deep Learning Model for Underwater Species Recognition in Aquaculture" introduces the FishDERECT Model Architecture developed by the authors. This model was trained using fish masks and led to significant improvements in fish detection accuracy. This innovation may be noteworthy in their region and merits publication in Fishes after undergoing a major revision.
Major points:
1. The manuscript does not directly relate to species recognition in aquaculture, as neither of the study cases involves aquaculture farms.
2. The structure of the manuscript should be revised to address confusing expressions (please refer to the detailed notes below).
3. Each tool used in the study, such as software, Python packages (libraries), and web applications, should be clearly defined.
4. Please review the accuracy of figure captions.
5. Verify the correct style for in-text references.
6. Ensure thorough proofreading for language accuracy, particularly regarding capitalized letters and incomplete sentences.
Specific comments:
Line 39: I agree that manually detecting fish can be harder due to the complex background. But, the results with Gilt Head Bream also had issues with the background. Please explain how the method in this study compares to manual detection or other automatic methods?
Line 41: Change "backdrop" to "background."
Line 49: Start a new paragraph from "In this paper..."
Line 124: Is the CarMask dataset about car images? We need more details about the CarMask dataset. If it's open, please provide a link. If not, maybe show some samples (and include them in the Supplementary Materials).
Line 126: Change "receive" to "receives."
Line 132: There's a grammar issue with "Used to..."
Line 136: I'm sorry, but I got confused here. The FishMask dataset should already include predicted fish masks (from L133-134), so why use Roboflow to label something? I later found the answers in L156-158. Could you reorganize this part?
Line 140: U-net might be a type of FCN. Check and clarify.
Line 150: What is the CarMask model? (although I got the answer from Sect. 3.3.1). What images are you referring to?
Line 159: What is "transfer learning"? I found the answers in L213-216. Please restructure this part.
Line 181: How were the weights determined?
Line 183: What is the "Species Recognition Model"?
Line 186: Sorry, but I don't understand "constrained performances device."
Line 188: I think the "embedded AI model" is the FishDETECT model. Can you put it more simply?
Line 189: Why is "Camera" capitalized?
Line 189: Maybe "innovative" isn't right considering global AI development. Remove it or add constraints like "locally" or "nationwide," depending on AI's development in the region.
Line 190: "aquatic lifeforms" might be better as "various fish species," like in L195.
Line 193: "visual data" could be "images," as in L196.
Line 197: Change "species present" to "present species."
Line 206: What do you mean by "various format"? Include some training dataset samples.
Line 224: Remove the sentence "It is called fishMask model in the following of this paper. it consists in training on the 224 FishMask dataset containing the images of fish masks."
Line 231: "mAP50"
Line 228: It's TABLE 1, not TABLE I.
Line 242: There might be something missing in Sect. 4.2.
Line 251: I'm curious why shrimp is the simplest task. Maybe provide a sample image.
Line 255: "Black Sea Sprat"?
Line 260: If images were analyzed for the embedded device, were they taken from real-time videos? Because "In the test phase, underwater videos were captured under different conditions." If so, you can extract many images and provide performance metrics, e.g., show average detection rate and standard deviation under similar conditions for clarity.
Line 272: "TABLE 2"
Ensure thorough proofreading for language accuracy, particularly regarding capitalized letters and incomplete sentences.
Reviewer 2 Report
The authors investigated an improved deep learning model for underwater species recognition in aquaculture. This manuscript (MS) was clearly written and easy to understand. The topic is novel way and interesting how in the future, this method can help aquaculture. The only comment I have is to improve the English of MS and fix errors.
Few errors to improve.
Author Response
We have worked on english writing.
Reviewer 3 Report
Thank you for giving me the opportunity to review `An Improved Deep
Learning Model for Underwater Species Recognition in Aquaculture`
In this paper the authors present a model based on YoloV5 which has used
transfer learning to retrain the model on features extracted from a fish
species data set using U-Net.
The novelty of this study is not very well explained. Data sets are
introduced with insufficient justification. If I have understood the
study correctly, The authors have a data set of fish that do not have
the extent of the fish labelled sufficiently for training a YOLO based
object detector. Therefore, they first segment the fish using unet which
will describe the extent of the fish or multiple fish. YOLO requires
bounding boxes to train. These boxes could in principle be used to
train the fish detector but the step for generating a bounding box from
a mask is omitted. There are data sets pre-labelled with bounding boxes
specifically for training YOLO such as
<https://github.com/open-AIMS/ozfish> . Why not use that? There may be
some novelty for generating bounding boxes from unlabelled fish in order
to increase data set sizes, but the authors have not described this as
their goal
Perhaps the most the biggest issue I believe in this manuscript is, the
Authors have done nothing to address the generalisability of their
model. There is no mention of in-sample data splits between training,
validation and test sets an no quantitative measure of out-of-sample
testing. With a confusion matrix that suggests 100% accuracy of most
species of fish really suggests that the model is over-fit and that the
authors are trying to present the results from their training data not test. Over
fitting is a big problem for machine learning models, and needs to be
addressed by the authors. Where is this model going to be used ? what
is its application. If the authors are suggesting they can reproduce
these results in realworld unconstrained environments, then they have
not presented evidence of it. Instead if the authors are mearly using
this data set to highlight novelty of the model itself with improvements
to object detection I do not feel training YOLO on fish4kknowlege is
novel enough by itself, even with the edition of generating masks from
unlabelled images. What happens if the model is presented a species of
fish it has not seen in its training data ? If this model is going to
be presented to underwater images of fish, it will indeed have to deal
with this scenario.
The authors are also pretty light on references which use deep learning
for fish species identification both YOLO and Mask-RCNN's have been used
in a number of recent studies omitted from the literature review.
In summary, there maybe some novel work here if the story of the paper
is made more clear. I suggest that the authors address the literature
more critically and highlight what this paper addresses that others have
not. Perhaps there is a story around generating bounding boxes from
unlabelled images which may add to the corpus of available data for
training .. perhaps. There is literature published that addresses this
already. If the authors are presenting their results from their training
data and have not done anything to mitigate over-fitting the model, I
find it difficult to justify publication.
1 General
═════════
There are missing table numbers and formatting errors that need to be
fixed up through out the manuscript. Descriptions of the datasets
including number of images, number of labels and label distribution
are insufficient. Scientific writing uses words like "our results
were very good" This needs to be addressed throughout the manuscript
and would benefit from a scientific editor.
2 Introduction
══════════════
3 Related work
══════════════
I feel this section needs to be better organised, it reads like its
just a long list of studies and there isn't enough critical writing
here. What were the advancements on the more recent papers, what were
the weaknesses or remaining challenges with these studies. If the
authors are suggesting that there are advancements with each of the
studies listed, its not clear to the reader. Perhaps some subheadings
that guide the reader indicating what these advancements are or can be
grouped into. Perhaps the evolution of computer vision methods like
bag of words, to CNNs to deep networks and more recently single-shot
object detectors. Somethings that tells a story rather than a list of
previous work.
[l-64] Vikram Deep & al. Should be et. al.
[l-65] 'famous' ? I would drop that kind of language. I don't know
what this particular journals stance is on citing data sources, but
perhaps fish4knowledge should include
<https://homepages.inf.ed.ac.uk/rbf/Fish4Knowledge/>.
[l-69] The sentence "After training the neural network, a model for
fish species recognizing from dimensions was generated." Doesn't make
sense to me. I don't understand 'recognizing from dimensions'
[l-104] Its not essential, YOLO doesn't do segmentation but it does
do object detection
[l-112] "The results of object detection using
this method are generally good" generally good is not scientific
language. Comparative quantitative language should be used. More
accurate that X or faster inference speed than Y
[l-112] "Many studies have used YOLO to solve problems related to
object detection in computer vision" … such as ? this seems a bit
lazy, why are these studies relevant to this particular manuscript.?
[l-106] What problems specifically. The authors haven't even
explained what problems are being addressed or being solved.
4 Methodology
═════════════
4.1 Data Gathering.
───────────────────
Where does CarMask come from. I cant find it or any reference to it.
What makes up this data set, why is it appropriate for this study, why
did the authors chose it ?
4.2 Data Preprocessing.
───────────────────────
What is Roboflow, I guess the authors are referring to the online
platform but its not that clear. https://roboflow.com/
4.3 CarMask Model
─────────────────
This model is not described in any detail or really justified for
inclusion.
5 Result and Discussion
═══════════════════════
Figure 6. This does not show the confusion matrix, it shows a single
example of the detection results. This lack of attention to detail is
throughout the manuscript.
6 Further Discussion
════════════════════
[l-276] There are a number of studies addressing changes in habitats
<https://pubmed.ncbi.nlm.nih.gov/33044609/> and changes in water
quality.
7 References
════════════
[l-350] 'In2019' format reference
The Authors use language such as ""our results were very good". The manuscript needs to use quantitative and / or comparative language.
Round 2
Reviewer 1 Report
Thank you for the diligent effort put into the revision. I acknowledge that the authors may not fully concur with all the comments, but I appreciate their earnest attempts to enhance the manuscript.
Regarding the topic, it is indeed unnecessary to acquire images from aquaculture farms. However, my primary concern lies more in the practical application of this technology within aquaculture environments, which can often present challenging conditions for image recognition (e.g., poor water quality and high fish density). If the authors' understanding of aquaculture farms is based on the content shared via their provided link (https://www.youtube.com/watch?v=Wm4Js_ZKrM0&t=2s), then I wholeheartedly agree that their approach is applicable in such contexts.
Regarding language, I apologize if my previous comments were inadvertently offensive. I was somewhat surprised by the language quality in the initial submission, given that we are in an era where computers can generate text that is not only coherent but also largely free from grammatical errors. Additionally, the authors are presumably well-versed in the relevant technology. Please accept my apologies for any unintended bluntness. I must commend the authors for the significant improvement in the quality of English in this version.
In terms of content, I have noticed the addition of significantly more details in this version, rendering it much more comprehensible.
However, with regards to the format, I recommend a thorough review of the references. For instance, are references #10 and #16 referring to the same source? It's important to understand the necessity of each reference and ensure appropriate citations when required. Regrettably, due to time constraints, I am unable to conduct a comprehensive reference check myself. This is the only comment I have for this review. Thank you.
Author Response
However, with regards to the format, I recommend a thorough review of the references. For instance, are references #10 and #16 referring to the same source? It's important to understand the necessity of each reference and ensure appropriate citations when required. Regrettably, due to time constraints, I am unable to conduct a comprehensive reference check myself. This is the only comment I have for this review. Thank you.
All references have been revised. References 10 and 16 represent the same source. Reference 16 has been removed